# Comparison of OneChoice AI-based clinical decision support recommendations with infectious disease specialists and non-specialists for bacteremia treatment in Lima, Peru

Juan Carlos Gómez de la Torre[1,2,3,4☉], Ari Frenkel[2☉], Carlos Chavez-Lencinas[2,5,6☉], Alicia Rendon[2☉], Max Fabian[2☉], Jose Alonso Caceres-DelAguila[1☉], Miguel Hueda-Zavaleta[2,7]*

1 Clinical Laboratory Roe, Lima, Peru, 2 Arkstone Medical Solutions, Boca Raton, Florida, United States of America, 3 Instituto de investigaciones en ciencias biomedicas, Universidad Ricardo Palma, Lima, Peru, 4 Facultad de Medicina, Universidad Ricardo Palma, Lima, Peru, 5 Hospital Nacional Edgardo Rebagliati Martins, Lima, Peru, 6 Facultad de Medicina, Universidad Nacional Mayor de San Marcos, Lima, Peru, 7 Facultad de Ciencias de la Salud, Universidad Privada de Tacna, Tacna, Peru

☉ These authors contributed equally to this work.
* mighueda@virtual.upt.pe

## Abstract

Bacteremia is a major contributor to global morbidity and mortality, particularly in low- and middle-income countries where diagnostic delays and empirical antimicrobial misuse exacerbate resistance. This study assessed the accuracy of OneChoice®, an artificial intelligence (AI)-based Clinical Decision Support System (CDSS), in guiding antimicrobial therapy for bloodstream infections (BSIs) in Lima, Peru. A cross-sectional, observational design was used, comparing therapeutic recommendations generated by OneChoice®—based on molecular (FilmArray®) and phenotypic (MALDI-TOF MS, VITEK2) data—with the clinical decisions of 94 physicians (35 infectious disease [ID] specialists and 59 non-specialists) across 366 survey-based evaluations of bacteremia cases. Concordance between CDSS and physician decisions was analyzed using Cohen's Kappa and logistic regression. The overall concordance rate was 96.1% when considering any suggested treatment, and 74.6% for the top recommendation, with a substantial agreement (κ = 0.70). ID specialists showed significantly higher concordance (κ = 0.78) than non-ID physicians (κ = 0.61), and specialization was the strongest predictor of agreement (OR = 2.26, p = 0.001). Escherichia coli cases had the highest concordance, while Pseudomonas aeruginosa showed the lowest. The CDSS reduced inappropriate antibiotic use, particularly unnecessary carbapenem prescriptions. These findings support the utility of AI-CDSS tools in enhancing antimicrobial stewardship and standardizing care, especially in resource-limited healthcare settings.

**Data availability statement:** All relevant data are within the paper and its Supporting information files. In addition, the document of 'Supp 3.pdf' has been uploaded to https://doi.org/10.6084/m9.figshare.31281952 to ensure permanent accessibility.

**Funding:** The author(s) received no specific funding for this work.

**Competing interests:** Ari Frenkel is Chief Science Officer of Arkstone Medical Solutions, the company that produces the OneChoice report evaluated in this study. JC Gómez de la Torre works as the Director of Molecular Informatics at Arkstone Medical Solutions and as the Medical Director at Roe Lab in Perú. At the same time, Alicia Rendon and Miguel Hueda Zavaleta serve as Quality Assurance Managers at Arkstone Medical Solutions. These affiliations may be perceived as potential conflicts of interest. However, the study's design, data collection, analysis, interpretation, manuscript preparation, and the decision to publish the results were conducted independently, with no undue influence from the authors' affiliations or roles within the company.

## 1. Introduction

Sepsis and bloodstream infections (BSIs) represent a critical global health issue, particularly in low- and middle-income countries (LMICs), where delayed diagnoses and treatment gaps compound mortality [1]. In 2020 alone, an estimated 11 million deaths globally were attributed to sepsis, making it one of the leading causes of death worldwide [2]. In Latin America, limited diagnostic capacity and inappropriate antibiotic use exacerbate the crisis, leading to increasing multidrug-resistant (MDR) infections [2].

Rapid and accurate diagnosis is essential for effective management, as delays in appropriate antimicrobial therapy have been associated with increased mortality [3]. Traditional diagnostic methods, such as blood cultures and phenotypic antimicrobial susceptibility testing (AST), remain the gold standard but suffer from long turnaround times, typically requiring 24–72 h for actionable results [4–6]. This diagnostic delay is directly associated with increased mortality and inappropriate empirical therapy [6].

BSIs caused by organisms like *Escherichia coli*, *Klebsiella pneumoniae*, and *Pseudomonas aeruginosa* are increasingly resistant to standard antibiotics, driving the need for faster and more precise diagnostic tools [7]. Recent advances in molecular diagnostics, including FilmArray® (BioFire Diagnostics, LLC, Salt Lake City, UT, USA) and GeneXpert® (Cepheid, Sunnyvale, CA, USA), have significantly reduced this time from days to hours, allowing for faster pathogen identification and resistance profiling [8]. However, effectively integrating such diagnostic tools into clinical workflows remains challenging, necessitating more robust decision support mechanisms.

When integrated with artificial intelligence (AI), these tools form the backbone of clinical decision support systems (CDSSs) and can provide personalized antimicrobial recommendations [9]. AI-based CDSSs, such as OneChoice® developed by Arkstone, are particularly promising for infectious disease management because they synthesize vast datasets in real time and align with clinical guidelines. This AI-CDSS integrates molecular (e.g., FilmArray) and phenotypic (e.g., VITEK2 and MALDI-TOF MS) diagnostic outputs to offer antimicrobial treatment recommendations. Its design aligns with global and regional stewardship goals, including those from the Surviving Sepsis Campaign and IDSA [10,11]. The OneChoice® CDSS, powered by a machine learning model, has demonstrated strong internal validity in antimicrobial stewardship. In one study that examined the accuracy of its internal validation, the model revealed that it can accurately distinguish trained from novel data (100.0% accuracy) and showed 84.0% agreement with clinical standards in minor discrepancies while avoiding major discrepancies entirely. These results, confirmed through multiple validation methods and enhanced by human-in-the-loop (HITL) oversight, support the system's potential as a reliable tool for optimizing antimicrobial therapy [12].

However, OneChoice® is currently classified as a clinical decision support tool and has not yet received formal regulatory approval as a medical device from the U.S. Food and Drug Administration (FDA) or CE marking under European Union Medical Device Regulation. The system is intended to assist clinical decision-making and does not replace physician judgment. The recommendations generated should be interpreted within the context of individual patient characteristics and local guidelines, and the tool should be used under appropriate clinical supervision.

These systems have demonstrated potential in reducing inappropriate antibiotic use, optimizing dosing, and ensuring compliance with antimicrobial stewardship principles [13]. While initial results show promise in high concordance with expert ID physicians' recommendations, real-world validation remains limited, especially in settings such as Peru.

Artificial intelligence (AI) and machine learning (ML) advancements have revolutionized diagnostics in infectious diseases. AI models are increasingly integrated into CDSSs, providing real-time, tailored therapeutic recommendations based on molecular and phenotypic data [13]. Such systems have effectively reduced diagnostic turnaround times and improved clinical outcomes [14]. AI-CDSS applications include early sepsis detection, resistance prediction, and antimicrobial therapy optimization [15,16]. Moreover, in regions like Latin America, AI-driven CDSSs may help bridge technological gaps and support overburdened healthcare systems [17].

Given the complex interplay of clinical, microbiological, and pharmacological factors in BSI management, AI-driven CDSSs have the potential to standardize therapeutic approaches and minimize errors due to human variability. AI-enhanced platforms can analyze diverse datasets in real time, offering context-specific recommendations that align with stewardship principles [18]. OneChoice® is designed to mitigate the therapeutic lag between pathogen detection and treatment initiation, particularly critical in time-sensitive infections such as sepsis. OneChoice® has demonstrated that it can deliver rapid and accurate therapeutic recommendations in cases of bacteremia. An 80.0% concordance was observed between the recommendations generated from molecular and phenotypic results. Furthermore, molecular-based recommendations were available up to 29 h earlier than those based on phenotypic data, suggesting that integrating molecular diagnostics with AI-driven CDSSs could significantly accelerate clinical decision making in bacteremia. Notably, most discrepancies were due to the inability of certain molecular platforms to detect specific phenotypic resistance mechanisms [19].

Despite the promise of CDSS platforms, several barriers persist. These include clinician skepticism, system interoperability issues, and a lack of real-world validation [14,20]. Additionally, implementation in LMICs like Peru is hindered by limited infrastructure and low CDSS awareness among healthcare professionals [17].

Studies evaluating CDSSs such as OneChoice® remain limited, and few investigations have rigorously assessed its alignment with expert clinical judgment in LMICs, particularly among non-specialist physicians.

In Peru, a country with documented high resistance rates and an underfunded infectious disease infrastructure [17], the clinical validation of AI-CDSS tools is critically needed. This study is among the first to evaluate the agreement between an AI-CDSS and physicians across specialties, providing crucial insights into how AI can bridge knowledge gaps and enhance precision medicine. This study compares the CDSS recommendations with therapeutic decisions made by infectious disease (ID) specialists and non-ID specialists, assessing their concordance. The findings will help inform implementation frameworks, training needs, and policy recommendations for AI's ethical and scalable deployment in infectious disease care.

## 2. Materials and methods

### 2.1. Study design and setting

This study employed a cross-sectional, observational design, conducted in tertiary healthcare facilities in Lima, Peru. The study utilized molecular and phenotypic diagnostic methods routinely applied to guide antimicrobial therapy. The primary objective was to compare the CDSS recommendations with therapeutic decisions made by infectious disease (ID) specialists and non-ID specialists, assessing their concordance. Confirmed bacteremia cases identified between January and December 2024 were included. These cases were diagnosed using routine molecular and phenotypic methods in clinical practice, with results available from both OneChoice® and OneChoice® Fusion reports. Demographic, clinical, microbiological, and CDSS-related data were collected between June 15 and June 25, 2025. Subsequently, electronic surveys were administered to infectious disease (ID) specialists and non-ID specialists from June 26 to July 15, 2025.

## 2.2. Participants and selection criteria

The study involved 94 physicians, comprising 35 infectious disease specialists and 59 non-ID specialists. Participants were selected based on their involvement in bacteremia management within the healthcare facilities. The inclusion criteria mandated that participating physicians had at least three years of clinical experience in infectious disease management. The study included confirmed bacteremia cases, selected based on strict inclusion criteria to ensure the reliability of the dataset. Eligible cases involved patients with positive blood cultures confirmed through FilmArray® molecular testing and phenotypic identification using MALDI-TOF MS and VITEK2 systems. Cases were excluded if they had incomplete laboratory data, invalid susceptibility results, or were determined to be contaminants rather than true BSIs.

## 2.3. Data acquisition and characteristics

Data for the study were acquired from confirmed bacteremia cases, integrating microbiological, clinical, and therapeutic information. The dataset included results from molecular diagnostics using the FilmArray Blood Culture Identification (BCID) Panel (BioFire Diagnostics, LLC, Salt Lake City, UT, USA) and phenotypic antimicrobial susceptibility tests conducted with Matrix-assisted laser desorption ionisation time-of-flight (MALDI-TOF MS) (bioMérieux, Inc. France) and VITEK2 system (bioMérieux, Inc. France). The initial OneChoice® report (S1 Appendix) was generated based on data from molecular results alone, while the final OneChoice® (Fusion) report (S2 Appendix) incorporated both molecular and phenotypic test results.

The study focused on bacterial BSIs, selecting cases with available molecular and phenotypic testing results. Clinical case surveys were administered to physicians, presenting cases twice, initially with molecular results alone, and later with combined molecular and phenotypic data. Responses were collected to evaluate the agreement between physicians and OneChoice®. In total, 366 case evaluations were collected, allowing for a direct comparison of CDSS recommendations versus clinical judgment (S3 Appendix). All data were fully anonymized before analysis, and authors had no access to identifying patient information during or after data collection.

## 2.4. Study procedures and tools/Instruments/Materials/Equipment

The experimental procedures followed a structured approach to ensure consistency and reproducibility. Blood samples from patients with suspected bacteremia were processed using the FilmArray BCID Panel, which operates under specific conditions: samples were incubated at 37°C for 24 hours before analysis. The MALDI-TOF MS was calibrated daily using standard bacterial strains to ensure accuracy in phenotypic identification. The VITEK2 system was employed for susceptibility testing, using standard inoculum sizes and incubation periods as per manufacturer guidelines.

Each physician received clinical case surveys electronically, with cases presented randomly to minimize bias. Surveys were designed using Microsoft Forms, ensuring uniformity in data collection (S3 Appendix). The CDSS utilized proprietary algorithms to generate therapeutic recommendations, which were then compared to those of physicians.

## 2.5. Data preparation

Before analysis, all collected data were anonymized to protect patient confidentiality. Data entry was performed using Microsoft Excel, with each case assigned a unique numeric code. The dataset underwent rigorous cleaning procedures to ensure accuracy, including validation checks for duplicate entries and cross-referencing against laboratory records to confirm diagnostic results (S4 Appendix).

In this study, concordance was defined as agreement between the therapeutic recommendation provided by the CDSS and the expert review based on molecular and/or phenotypic data by the Infectious and Tropical Disease and Non-Infectious Disease Specialists. Discordance refers to any disagreement with expert judgment regarding antimicrobial selection, dosing, or spectrum. Discordant cases were further categorized by type of mismatch, including incorrect

antibiotic choice; incorrect dosing or interval; both incorrect antibiotic choice and dosing; unnecessary broad-spectrum coverage; unnecessary broad-spectrum with carbapenem overuse; or a combination of these. Multidrug-resistant (MDR) organisms were defined as non-susceptible to at least one agent in three or more antimicrobial categories, while extensively drug-resistant (XDR) organisms were non-susceptible to all but one or two categories, according to international consensus definitions [21].

### 2.6. Data analysis

Statistical analyses were conducted using STATA v.17. Descriptive statistics summarized physician responses, concordance rates, and frequency distributions of bacterial species and resistance genes. Cohen's Kappa coefficient (κ) assessed inter-rater agreement, with interpretations based on established scales for agreement levels. Chi-square (χ²) tests examined associations between categorical variables, such as specialty, experience level, and concordance rates. Fisher's exact test was applied for bacteria-associated bacteremia cases with limited sample sizes.

Multivariate logistic regression identified independent predictors of concordance, adjusting for specialty, experience, and antimicrobial resistance status. Model specifications included interaction terms to explore the influence of combined factors on therapeutic decision-making.

To assess the concordance between the CDSS and the clinical judgment of the participants, each therapeutic recommendation generated by the CDSS was independently reviewed by two infectious disease specialists (JCGdIT and CChL). Cases were classified as concordant or discordant depending on whether the participant's decision aligned with the CDSS recommendation. For discordant cases, an adjudication process was conducted to determine whether the discrepancy favored the CDSS, the participant, or both. When the two infectious disease specialists disagreed in their classifications, a third specialist (MH-Z) evaluated the case, and a consensus decision was reached. In addition, all discrepancies were categorized into five mutually exclusive groups: incorrect antibiotic choice; combined errors in antibiotic selection and dosing or duration; unnecessary use of broad-spectrum carbapenems; unnecessary use of broad-spectrum antibiotics only; and discrepancies attributable to both perspectives.

This study was approved by the Institutional Ethics Committee of the Universidad Privada de Tacna, Peru (FACSA-CEI/093-06-2025). Informed consent was obtained from all participating physicians prior to survey administration, after being informed about the purpose of the study, the voluntary nature of their participation, confidentiality safeguards, and their right to withdraw at any time. The consent process was documented electronically through Microsoft Forms. Cases of bacteremia and associated microbiological data were derived from routine clinical care performed between January and December 2024, originally collected as part of standard clinical practice for diagnostic and therapeutic purposes. For the secondary use of these pre-existing clinical data in research, the Ethics Committee granted a waiver of informed consent based on the following criteria: (1) the research involved no more than minimal risk to participants; (2) the waiver did not adversely affect the rights and welfare of the participants; (3) the research could not practicably be carried out without the waiver; and (4) all data were fully anonymized prior to analysis, with patient names replaced by unique numerical codes to prevent identification. The study adhered to international ethical standards, ensuring confidentiality, data integrity, and research reproducibility. The findings will be submitted for peer-reviewed publication and presented at scientific conferences to promote awareness of CDSS integration in antimicrobial stewardship.

## 3. Results

A total of 366 bacteremia cases were evaluated, of which 206 (56.3%) were reviewed by infectious disease (ID) specialists and 160 (43.7%) by non-infectious disease (non-ID) physicians. A significant difference was observed in years of experience between the two groups (p = 0.001); 71.2% of ID specialists had between 1 and 5 years of experience, whereas 52.0% of non-ID physicians had between 16 and 20 years. The most frequently isolated bacterial species were *Escherichia coli* (35.5%), *Klebsiella pneumoniae/Klebsiella aerogenes* (15.8%), and *Salmonella typhi/enterica* (13.7%).

*Klebsiella spp.* and *Salmonella spp.* were more commonly observed in cases evaluated by ID specialists (65.6% and 64.0%, respectively). Additionally, *Serratia marcescens* was exclusively identified in cases reviewed by ID specialists. Regarding resistance genes, 59.0% of isolates showed no detectable resistance genes, with a significant difference between groups (p = 0.047). ID specialists more frequently evaluated bacteremia cases involving strains with *CTM-X* (61.3% vs. 38.7%) and *NDM* genes (60.0% vs. 40.0%). All cases with *VIM* beta-lactamase genes were reported in the non-ID group. Overall, 39.9% of isolates were classified as multidrug-resistant, with no significant differences between groups (p = 0.640) (Table 1).

In terms of concordance between the system's recommendations and expert evaluation, 74.6% of cases showed overall agreement, with a significantly higher rate in the ID group (61.2% vs. 38.8%, p = 0.001). Disagreement was more common among non-ID physicians (58.1% vs. 41.9%) (Table 1). Finally, A detailed assessment of discrepancies in therapeutic decisions revealed that 92.5% of discordant cases favored the CDSS recommendations, while 7.5% reflected consideration of both CDSS and physician input. The most frequent type of mismatch was incorrect antibiotic selection (48.4%), followed by combined errors in antibiotic choice, dosing, or interval (19.4%), and unnecessary broad-spectrum use (8.6%). Additionally, 10.8% of mismatches involved unnecessary carbapenem overuse, underscoring the importance of carbapenem stewardship in the context of resistance. Notably, most dosing-related errors occurred in the non-ID group (Table 1, Fig 1).

The overall concordance rate between OneChoice® recommendations and physician decisions was 96.1% when considering any of the four alternative treatment options provided by the system. Concordance was slightly lower when evaluating the best-recommended option, with rates of 74.3% for molecular-based results and 74.8% for phenotypic results. The Kappa Index, a statistical measure of agreement beyond chance, was calculated for all participants and stratified by specialty, the overall Kappa Index was 0.70, indicating substantial agreement. Infectious disease specialists demonstrated a higher concordance with a Kappa value of 0.78 than non-ID specialists, who had a Kappa value of 0.61 (Table 2).

The study further explored concordance rates based on physician experience, revealing higher agreement among more experienced physicians, particularly among ID specialists. Concordance rates among ID specialists ranged from 70.8% for those with 16–20 years of experience to 84.2% for those with more than 20 years. In contrast, non-ID specialists showed an overall lower concordance rate of 66.3%, with the highest agreement observed in the 16–20-year experience group at 80.7% (Table 3). The analysis of bacterial species and concordance rates revealed that Escherichia coli had the highest concordance rate at 80.0%. In contrast, Pseudomonas aeruginosa exhibited the lowest agreement at 53.8%, with a statistically significant p-value of 0.012. Conversely, Staphylococcus haemolyticus showed 100.0% concordance (Table 3).

A multivariate logistic regression analysis was performed to identify factors influencing concordance, adjusting for specialty, years of experience, and MDR status. The results, summarized in Table 4, indicated that being an ID specialist was the strongest predictor of concordance, with an odds ratio (OR) of 2.26 and a p-value of 0.001. Years of experience did not show a statistically significant effect on concordance, with an OR of 1.27 (p = 0.562) for physicians with 16–20 years of experience and an OR of 1.06 (p = 0.843) for those with more than 20 years. MDR status showed a trend towards higher concordance (OR = 1.52, p = 0.104), suggesting that physicians are more likely to follow CDSS recommendations when dealing with multidrug-resistant infections. However, this result was not statistically significant (Table 4).

Finally, we evaluated the overall level and nature of the discrepancies between the CDSS and the study participants. Concordance between the CDSS recommendations and the participants' assessments occurred in 74.6% of cases (273/366), while the remaining 25.4% showed discordance (Fig 2a). Among the discordant cases, the vast majority (92.5%) favored the CDSS (Onechoice) recommendation, while only 7.5% were considered acceptable by both parties (CDSS and expert reviewers) (Fig 2b). Likewise, the most frequent types of discrepancies were incorrect antibiotic choice (51.7%), combined errors in antibiotic selection, dosing, or duration (20.7%), unnecessary use of

**Table 1. Characteristics of evaluated bacteremia cases by specialist background: Microbial profiles, resistance genes, and concordance rates.**

| Variable | Total (n = 366) | Bacteremia cases evaluated by | | p-value |
| --- | --- | --- | --- | --- |
| | | Non-ID (n = 160) | ID (n = 206) | |
| Specialty Time (%) | | | | 0.001[a] |
| 1–5 years | 104 (28.4) | 30 (28.8) | 74 (71.2) | |
| 16–20 years | 50 (13.6) | 26 (52.0) | 24 (48.0) | |
| >20 years | 212 (58.0) | 104 (49.1) | 108 (50.9) | |
| Bacteria Type (%) | | | | 0.063[a] |
| Escherichia coli | 130 (35.5) | 66 (50.8) | 64 (49.2) | |
| Klebsiella pneumoniae/ aerogenes | 58 (15.8) | 20 (34.4) | 38 (65.6) | |
| Salmonella spp. | 50 (13.7) | 18 (36.0) | 28 (64.0) | |
| Streptococcus sp. | 34 (9.3) | 18 (53.0) | 16 (47.0) | |
| Pseudomonas aeruginosa | 26 (7.1) | 12 (46.2) | 14 (53.8) | |
| Enterococcus faecalis | 18 (4.9) | 6 (33.3) | 12 (66.7) | |
| Staphylococcus haemolyticus | 16 (4.4) | 6 (37.5) | 10 (62.5) | |
| Enterobacter sp. | 14 (3.8) | 8 (57.1) | 6 (42.9) | |
| Klebsiella oxytoca | 10 (2.7) | 6 (60.0) | 4 (40.0) | |
| Serratia marcescens | 10 (2.7) | 0 (0.0) | 10 (100.0) | |
| Resistance genes (%) | | | | 0.047[a] |
| No genes detected | 216 (59.0) | 90 (41.7) | 126 (58.3) | |
| CTX-M | 62 (16.9) | 24 (38.7) | 38 (61.3) | |
| ESBL | 56 (16.9) | 30 (53.6) | 26 (46.4) | |
| MecA/C | 16 (4.4) | 6 (37.5) | 10 (62.5) | |
| VIM beta-lactamase | 6 (1.6) | 6 (100.0) | 0 (0.0) | |
| NDM | 10 (2.7) | 4 (40.0) | 6 (60.0) | |
| Multidrug resistant (%) | | | | 0.640[a] |
| Multidrug resistant | 146 (39.9) | 66 (45.2) | 80 (54.8) | |
| Not Multidrug resistant | 220 (60.1) | 94 (42.7) | 126 (57.3) | |
| Concordance (%) | | | | 0.001[a] |
| Disagreement | 93 (25.4) | 54 (58.1) | 39 (41.9) | |
| Agreement | 273 (74.6) | 106 (38.8) | 167 (61.2) | |
| Mismatch Type (%) | | | | 0.001[a] |
| Incorrect Antibiotic Choice | 45 (48.4) | 29 (64.4) | 16 (35.6) | |
| Incorrect Antibiotic Choice, Incorrect Dosing or Interval | 18 (19.4) | 7 (38.9) | 11 (61.1) | |
| Incorrect Dosing or Interval | 6 (6.5) | 5 (83.3) | 1 (16.7) | |
| Unnecessary Broad Spectrum | 8 (8.6) | 4 (50.0) | 4 (50.0) | |
| Carbapenem overuse | 10 (10.8) | 7 (70.0) | 3 (30.0) | |
| More than one mismatch | 6 (6.5) | 2 (33.3) | 4 (66.7) | |

a: Chi-square; CTX-M: Cefotaximase-Munich; ESBL: extended spectrum beta-lactamase; NDM: New Delhi metallo-β-lactamases; VIM: Verona integron-encoded metallo-β-lactamases.

broad-spectrum or carbapenem agents (11.5%), unnecessary use of broad-spectrum antibiotics (9.2%), and discrepancies attributable to both perspectives (6.9%) (Fig 2c). These findings suggest that most discrepancies were due to inappropriate antibiotic selection and that when disagreement occurred, the CDSS recommendations were more accurate in most cases.

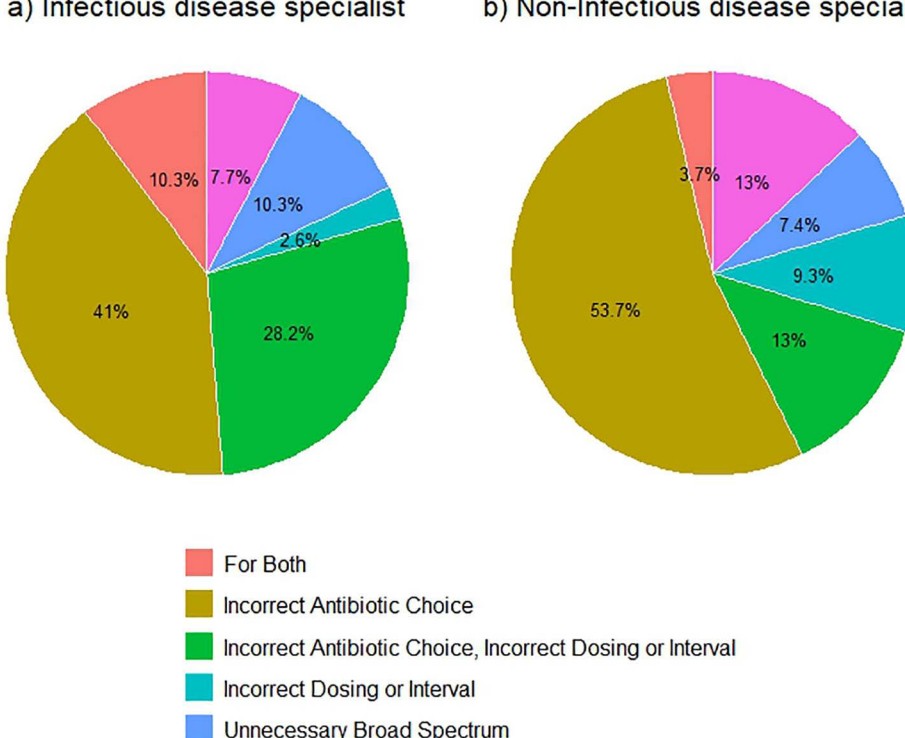

a) Infectious disease specialist    b) Non-Infectious disease specialist

Legend:
- For Both
- Incorrect Antibiotic Choice
- Incorrect Antibiotic Choice, Incorrect Dosing or Interval
- Incorrect Dosing or Interval
- Unnecessary Broad Spectrum
- Unnecessary Broad Spectrum, Carbapenem overuse

**Fig 1. Discrepancy type between Infectious disease specialist and Non-Infectious Disease Specialist and result type.**

**Table 2. Kappa Index regarding Antibiotic prescription between SSDD and Physicians.**

| Variable | Agreement | Kappa | p-value |
|---|---|---|---|
| All the participants | 74.6% | 0.7019 | 0.0000 |
| Infectious disease specialist | 81.0% | 0.7760 | 0.0000 |
| Non-Infectious Disease Specialist | 66.3% | 0.6079 | 0.0000 |

## 4. Discussion

This study evaluated the concordance between the therapeutic recommendations provided by an artificial intelligence–based clinical decision support system (CDSS), OneChoice®, and expert physician judgment in real-world bacteremia cases. Our findings demonstrate a substantial level of agreement (κ = 0.70), particularly among infectious disease specialists (κ = 0.78), suggesting that the system can offer clinically relevant support when interpreted by trained professionals. These results contribute to the growing body of evidence on the role of CDSS in the management of bloodstream infections (BSIs) and antimicrobial stewardship, particularly in high-resistance settings like Peru [22]. The consistently high antimicrobial resistance rates reported in national surveillance data underscore the urgent need for innovative, effective interventions [22,23]. The observed lower concordance among non-infectious disease specialists (κ = 0.61) further emphasizes the importance of clinical expertise and reinforces that such tools should serve as supervised decision aids rather than autonomous systems, especially in complex infectious disease contexts.

**Table 3. Frequencies of concordances and discordances in infectious disease and non-infectious disease doctors and type of discrepancies detected.**

| Variable | Concordances | Discordances | p-value |
|---|---|---|---|
| ID specialist by years | | | 0.295[a] |
| 0-5 years | 59 (79.7%) | 15 (20.3%) | |
| 16-20 years | 17 (70.8%) | 7 (29.2%) | |
| >20 years | 91 (84.3%) | 17 (15.7%) | |
| Non-ID specialist by years | | | 0.211[a] |
| 0-5 years | 20 (66.7%) | 10 (33.3%) | |
| 16-20 years | 21(80.8%) | 5 (19.2%) | |
| >20 years | 69 (66.3%) | 35 (33.7%) | |
| Type of bacteria (Most common) | | | |
| Escherichia coli | 104 (80.0) | 26 (20.0) | 0.078[a] |
| Klebsiella pneumoniae | 31 (70.4) | 13 (29.6) | 0.502 [a] |
| Pseudomonas aeruginosa | 14 (53.8) | 12 (46.2) | 0.012 [a] |
| Salmonella spp. | 36 (72.0) | 14 (28.0) | 0.651 [a] |
| Staphylococcus haemolyticus | 16 (100.0) | 0 (0.0) | 0.015[b] |
| Detection of Resistance Genes | | | 0.212 [a] |
| Resistance genes detected | 117 (78.0%) | 33 (22.0%) | |
| No genes detected | 156 (72.2%) | 60 (27.8%) | |
| Phenotype of Resistance | | | 0.135 [a] |
| MDR | 115 (78.8%) | 31 (21.2%) | |
| Non MDR | 158 (71.8%) | 62 (28.2%) | |

a: Chi-square; b: Fisher exact test, ID: infectious disease, MDR: multidrug resistant.

**Table 4. Multivariate Logistic Regression Analysis of Factors Associated with Concordance Between Physician Decisions and OneChoice® Recommendations.**

| Concordance | OR | p value | 95% CI |
|---|---|---|---|
| Specialty | | | |
| No infectious disease specialist | Reference | | |
| Infectious disease specialist | 2.26 | **0.001** | 1.38 - 3.68 |
| Years of specialty | | | |
| 0 - 5 years | Reference | | |
| 16-20 years | 1.27 | 0.562 | 0.56 - 2.87 |
| > 20 years | 1.06 | 0.843 | 0.60 - 1.86 |
| Phenotype of Resistance | | | |
| Non MDR | Reference | | |
| MDR | 1.51 | 0.104 | 0.917 - 2.50 |

OR: odds ratio; CI: confidence interval; MDR: multidrug resistant.

Our results demonstrate a high concordance (96.1%) between the CDSS used and physician-prescribed treatments, reflecting the potential for AI-driven tools to support antimicrobial decision-making in real-world settings [16]. Substantial agreement (Kappa Index = 0.70) further supports its reliability. These findings align with existing literature highlighting the utility of machine learning-based systems in infectious disease management [14,15].

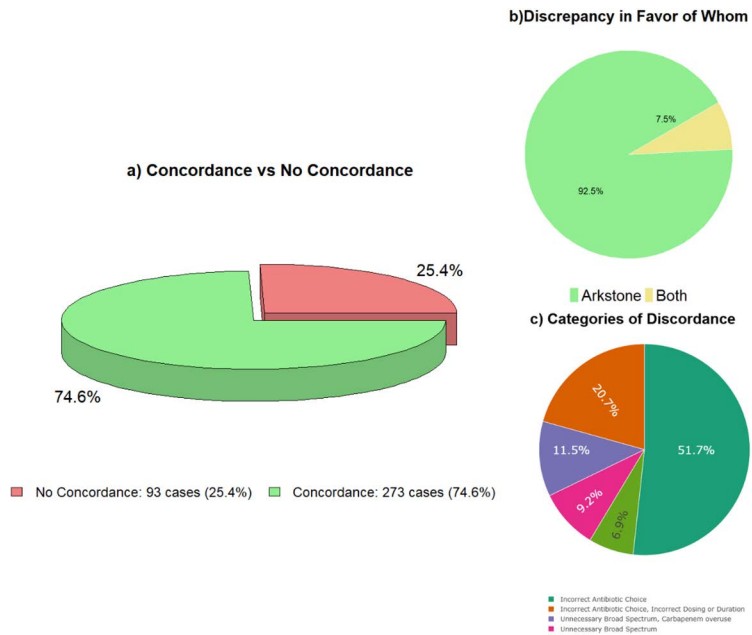

**Fig 2. Discordance between CDSS and participants.**

This study is among the first to evaluate the accuracy of the OneChoice® CDSS in a Peruvian clinical setting and across physician specializations. Higher concordance rates among ID specialists (Kappa = 0.78) compared to non-ID physicians (Kappa = 0.61) highlight the benefit of domain-specific expertise in interpreting CDSS outputs [16]. AI tools such as OneChoice® may be instrumental in bridging the knowledge gap among non-specialist practitioners [14].

Logistic regression identified ID specialization as the primary predictor of concordance (OR = 2.26, p = 0.001), reinforcing the role of targeted training. Interestingly, years of clinical experience showed no significant effect on concordance, suggesting expertise specificity is more relevant than time in practice.

Moreover, the system's ability to prevent unnecessary carbapenem use in 10.8% of discordant cases aligns with stewardship goals and can reduce selective pressures leading to resistance [13]. Nathan D. Nielsen et al. have noted that CDSS can improve antibiotic appropriateness in critical care settings [24].

This CDSS demonstrated particular strength in cases involving multidrug-resistant (MDR) organisms, such as Pseudomonas aeruginosa, where therapeutic decisions are inherently complex. Notably, P. aeruginosa had the lowest concordance rate (53.8%, p = 0.012), highlighting the critical need for structured tools in guiding antibiotic selection against resistant pathogens [25]. This aligns with prior studies emphasizing that AI-based diagnostics can enhance early pathogen recognition and streamline antimicrobial therapy [26].

While logistic regression analyses identified ID specialization as the strongest predictor of concordance (OR = 2.26, p = 0.001), years of clinical experience did not significantly influence agreement rates. This suggests that expertise in infectious diseases may play a more critical role in aligning with AI-generated recommendations than overall experience [26].

In contrast to the study by Montiel-Romero et al., which reported moderate concordance (κ = 0.48) in antibiotic selection and low concordance (κ = 0.39) in identifying resistance mechanisms between ChatGPT® and infectious disease (ID) specialists in simulated clinical cases [27], our study evaluated an artificial intelligence–based clinical decision support system (OneChoice®) applied to real-world bacteremia cases. We found a higher level of agreement (κ = 0.70) between

the system's recommendations and physician evaluations. Moreover, concordance was significantly higher among ID specialists ($\kappa = 0.78$) compared to non-specialists ($\kappa = 0.61$). These findings suggest that, unlike general language models such as ChatGPT®, systems specifically trained on structured clinical data may provide more reliable support in real clinical settings. Nevertheless, consistent with the conclusions of Montiel-Romero et al., our results underscore the importance of using these tools as supervised support rather than as a replacement for clinical judgment, particularly in scenarios where microbiological and pharmacological interpretation is critical for decision-making.

Despite these promising results, the discussion must acknowledge certain limitations. First, the sample size (366 physician surveys) and cross-sectional design limit generalizability and introduce potential confounders unaccounted for in regression modeling. Reliance on survey responses may introduce subjective bias, particularly if treatment deviated from guidelines. Second, our findings are specific to bacteremia and may not extend to non-bacteremic infections such as respiratory, urinary tract, or intra-abdominal syndromes. The real-world impact of this CDSS should be interpreted within the context of bloodstream infections. Third, we focused on concordance as a process metric rather than patient-centered outcomes such as mortality, length of stay, or clinical improvement. While substantial agreement with specialist judgment demonstrates clinical relevance, it does not directly prove improved patient outcomes. Fourth, findings from this Peruvian tertiary care setting may not generalize to healthcare systems with different diagnostic infrastructure, stewardship program maturity, or clinical workflows. Finally, lower concordance among non-specialists ($\kappa = 0.61$) highlights risks of overreliance on AI recommendations without adequate clinical reasoning, particularly in complex scenarios or when input data are incomplete.

Our findings demonstrate that the CDSS developed by Arkstone provides reliable therapeutic recommendations that align closely with clinical decision-making, particularly among infectious disease specialists. The substantial overall concordance and Kappa Index values indicate that the CDSS is a valuable tool for guiding antimicrobial therapy in bacteremia cases. The higher concordance rates observed among ID specialists and more experienced physicians suggest that specialized training and clinical expertise enhance alignment with CDSS recommendations. Furthermore, the detailed analysis of discrepancies highlights specific areas where the CDSS can aid in optimizing antibiotic selection and dosing, thereby contributing to improved antimicrobial stewardship.

Overall, this CDSS shows significant promise in aligning therapeutic recommendations with best practices, especially for common pathogens and experienced physicians. By reducing variation and enhancing evidence-based decision-making, such tools can support stewardship goals and improve care quality in resource-constrained settings.

AI-driven CDSS represent a promising advancement in infectious disease management. By enhancing diagnostic precision and standardizing antimicrobial therapy, these systems have the potential to mitigate the global burden of antimicrobial resistance and improve patient outcomes [28]. However, future CDSS iterations must incorporate clinical presentation, local epidemiology, host factors, and dynamic patient status to provide comprehensive support across diverse infectious syndromes beyond bacteremia. Future research should prioritize prospective, multicenter studies assessing CDSS impact on patient-centered outcomes, antimicrobial resistance trends, and healthcare resource utilization across diverse settings. Key directions include: (1) expansion of CDSS validation beyond bacteremia to encompass respiratory, urinary, and intra-abdominal infections; (2) integration of dynamic clinical data streams, local epidemiological inputs, and rapid molecular diagnostics such as FilmArray® BCID2 to enable context-sensitive decision support [29]; (3) incorporation of real-time resistance tracking and genomic insights to refine AI-CDSS applications; and (4) development of multidisciplinary implementation frameworks and training programs to ensure CDSS complements, rather than supplants, expert clinical judgment.

## 5. Conclusions

This study demonstrates that the OneChoice® AI-based clinical decision support system achieves substantial concordance ($\kappa = 0.70$) with physician therapeutic decisions in bacteremia management, with particularly strong agreement among

infectious disease specialists ($\kappa = 0.78$). The system showed a high overall concordance rate of 96.14% when considering any suggested treatment option, and 74.59% for the top recommendation. Infectious disease specialization was identified as the strongest predictor of concordance with CDSS recommendations (OR = 2.26, $p = 0.001$). The system demonstrated potential in reducing inappropriate antibiotic prescriptions, particularly unnecessary carbapenem use, supporting antimicrobial stewardship goals in a high-resistance setting. These findings support the utility of AI-CDSS tools as supervised decision aids for enhancing antimicrobial therapy standardization, especially in resource-limited healthcare settings.

## Supporting information

**S1 Appendix. Example of OneChoice® report based on molecular results alone.** https://doi.org/10.6084/m9.figshare.31281913.
(PDF)

**S2 Appendix. Example of OneChoice® Fusion report incorporating molecular and phenotypic data.** https://doi.org/10.6084/m9.figshare.31281946.
(PDF)

**S3 Appendix. Clinical case survey instrument used for physician–CDSS agreement analysis.** https://doi.org/10.6084/m9.figshare.31281952.
(PDF)

**S4 Appendix. Database.** https://doi.org/10.6084/m9.figshare.30994498.
(XLSX)

## Acknowledgments

We thank all the personnel in the Arkstone Medical Solutions and Roe clinical laboratory who have actively been working.

## Author contributions

**Conceptualization:** Juan Carlos Gómez de la Torre, Ari Frenkel.

**Data curation:** Juan Carlos Gómez de la Torre, Max Fabian, José Caceres.

**Formal analysis:** Juan Carlos Gómez de la Torre, José Caceres, Miguel Hueda-Zavaleta.

**Funding acquisition:** Juan Carlos Gómez de la Torre, Ari Frenkel.

**Investigation:** Juan Carlos Gómez de la Torre, Ari Frenkel, Alicia Rendon, Miguel Hueda-Zavaleta.

**Methodology:** Juan Carlos Gómez de la Torre, Carlos Chavez-Lencinas, Miguel Hueda-Zavaleta.

**Project administration:** Juan Carlos Gómez de la Torre, Ari Frenkel.

**Resources:** Juan Carlos Gómez de la Torre, Ari Frenkel, Carlos Chavez-Lencinas, Max Fabian, Miguel Hueda-Zavaleta.

**Software:** Juan Carlos Gómez de la Torre, Ari Frenkel, Max Fabian.

**Supervision:** Ari Frenkel, Alicia Rendon.

**Validation:** Juan Carlos Gómez de la Torre, Alicia Rendon, José Caceres, Miguel Hueda-Zavaleta.

**Visualization:** Juan Carlos Gómez de la Torre, Ari Frenkel, Max Fabian, José Caceres, Miguel Hueda-Zavaleta.

**Writing – original draft:** Juan Carlos Gómez de la Torre, Miguel Hueda-Zavaleta.

**Writing – review & editing:** Juan Carlos Gómez de la Torre, Ari Frenkel, Carlos Chavez-Lencinas, Alicia Rendon, Miguel Hueda-Zavaleta.

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
