## [Decision Letter · Decision Letter 0]

9 Dec 2025

Dear Dr. Hueda-Zavaleta,

Thank you for submitting your manuscript to PLOS ONE. After careful consideration, we feel that it has merit but does not fully meet PLOS ONE’s publication criteria as it currently stands. Therefore, we invite you to submit a revised version of the manuscript that addresses the points raised during the review process.

**ACADEMIC EDITOR'S COMMENTS:** Line 62-64: The authors stated "Traditional diagnostic methods, such as blood cultures and phenotypic antimicrobial susceptibility testing (AST), remain the gold standard but suffer from long turnaround times, typically requiring 24 to 72 h for actionable results. There are no references. More references should be cited, with this one (Zeng X, et al. 2025. Identification and characterization of an invasive, hyper-aerotolerant Campylobacter jejuni strain causing bacteremia in a pediatric leukemia patient. ASM Case Rep 1:e00060-24.

https://doi.org/10.1128/asmcr.00060-24) as an example (citing is optional).

We look forward to receiving your revised manuscript.

Kind regards,

Benjamin M. Liu, MBBS, PhD, D(ABMM), MB(ASCP)

Academic Editor

PLOS One

Journal Requirements:

“Ari Frenkel is Chief Science Officer of Arkstone Medical Solutions, the company that produces the OneChoice report evaluated in this study. JC Gómez de la Torre works as the Director of Molecular Informatics at Arkstone Medical Solutions and as the Medical Director at Roe Lab in Perú. At the same time, Alicia Rendon and Miguel Hueda Zavaleta serve as Quality Assurance Managers at Arkstone Medical Solutions. These affiliations may be perceived as potential conflicts of interest. However, the study's design, data collection, analysis, interpretation, manuscript preparation, and the decision to publish the results were conducted independently, with no undue influence from the authors’ affiliations or roles within the company.”

We note that one or more of the authors are employed by a commercial company: name of commercial company.

A. Please provide an amended Funding Statement declaring this commercial affiliation, as well as a statement regarding the Role of Funders in your study. If the funding organization did not play a role in the study design, data collection and analysis, decision to publish, or preparation of the manuscript and only provided financial support in the form of authors' salaries and/or research materials, please review your statements relating to the author contributions, and ensure you have specifically and accurately indicated the role(s) that these authors had in your study. You can update author roles in the Author Contributions section of the online submission form.

B. Please also provide an updated Competing Interests Statement declaring this commercial affiliation along with any other relevant declarations relating to employment, consultancy, patents, products in development, or marketed products, etc.

Reviewers' comments:

Reviewer's Responses to Questions

**Comments to the Author**

1. Is the manuscript technically sound, and do the data support the conclusions?

Reviewer #1: Partly

Reviewer #2: Yes

2. Has the statistical analysis been performed appropriately and rigorously?

Reviewer #1: Yes

Reviewer #2: Yes

3. Have the authors made all data underlying the findings in their manuscript fully available?

Reviewer #1: Yes

Reviewer #2: Yes

4. Is the manuscript presented in an intelligible fashion and written in standard English?

Reviewer #1: Yes

Reviewer #2: Yes

Reviewer #1: This paper presents a valuable contribution to the growing field of artificial intelligence-based clinical decision support systems (CDSS), focusing on the OneChoice® platform and its ability to guide antimicrobial therapy for bacteremia in a resource-limited setting. By comparing the recommendations of the CDSS with those of infectious disease (ID) specialists and non-specialists, the authors shed light on the potential of AI to standardize care and enhance stewardship however limited to the context of bloodstream infections.

Strengths of the Study

• Robust Methodology: The study employs a cross-sectional, observational design across tertiary healthcare facilities, integrating both molecular and phenotypic diagnostic data to ensure comprehensive evaluation.

• Expert Comparison: By including both ID specialists and non-specialists, the analysis highlights the impact of clinical expertise on concordance with CDSS recommendations, revealing substantial agreement, particularly among specialists (κ = 0.78).

• Quantitative Rigor: Statistical tools such as Cohen’s Kappa and logistic regression enrich the interpretation, and the use of a sizeable dataset (366 cases and 94 physicians) strengthens the findings.

• Emphasis on Antimicrobial Stewardship: The study demonstrates that the CDSS can help reduce inappropriate antibiotic use, especially carbapenem over-prescription, which is central to combating antimicrobial resistance.

Limitations and Critical Perspective

The following points could be addressed in the discussion:

• Narrow Clinical Focus: While bacteremia is a serious and high-impact condition, the paper’s exclusive focus on positive bloodstream infections means its findings apply only to a small minority of infectious disease patients. In clinical practice, most infections are not bacteremic, and the ability to act on blood culture results is relevant to a limited subset. Thus, the real-world impact of CDSS in guiding therapy for the broader population of infectious diseases may be overstated if conclusions are drawn solely from bacteremia cases.

• Scope of Clinical Decision Support: Effective decision support in infectious diseases must transcend the management of positive blood cultures. The spectrum of infectious syndromes—ranging from respiratory, urinary tract, and intra-abdominal infections to undifferentiated sepsis—requires nuanced recommendations that incorporate clinical presentation, local epidemiology, host factors, and dynamic changes in patient status. A system tailored only to bacteremia risks missing opportunities for intervention in far larger patient groups.

• Outcome Measures: The study centers on concordance rates rather than patient-centered outcomes, such as mortality, length of stay, or clinical improvement. While agreement with specialists is an important process metric, it does not guarantee improved patient outcomes. Future studies should measure the downstream effects of CDSS recommendations on clinical endpoints and stewardship goals.

• Generalizability: The context—a Peruvian tertiary care setting—limits extrapolation to different healthcare systems, especially where diagnostic infrastructure, stewardship programs, and clinical workflows differ.

• Potential for Overreliance: The authors rightly emphasize that CDSS should serve as supervised aids rather than autonomous systems. However, the risk remains that non-specialists may lean too heavily on AI recommendations without integrating broader clinical reasoning, especially in complex cases or where data inputs are incomplete.

• A glaring omission in the paper is a clear statement regarding the regulatory approval of the OneChoice® software tool. Nowhere in the manuscript is it made explicit whether the CDSS is authorized as a medical device under major regulatory frameworks such as the U.S. Food and Drug Administration (FDA) or has received a CE mark under European legislation.

Recommendations for Future Research

• Expand the scope of CDSS validation to include a wider array of infectious syndromes, incorporating real-world clinical scenarios beyond positive blood cultures.

• Assess the impact of CDSS on patient outcomes, antimicrobial resistance trends, and overall healthcare utilization, not merely process measures.

• Integrate dynamic clinical data streams and local epidemiological inputs, enabling the system to support nuanced decision-making across diverse infectious presentations.

• Promote multidisciplinary implementation and training to ensure that CDSS complements, rather than supplants, expert clinical judgment.

Conclusion

While the OneChoice® CDSS demonstrates substantial agreement with expert physicians in guiding therapy for bacteremia, its clinical effect is intrinsically limited by the small proportion of patients with positive blood cultures. The future of AI-driven decision support in infectious disease care lies in expanding beyond the narrow focus of bacteremia management to address the full complexity of infectious syndromes. Decision support platforms should be developed and validated to facilitate holistic, patient-centered, and context-sensitive care, ultimately enhancing outcomes for the broader population affected by infectious diseases.

Reviewer #2: AI-based clinical decision support systems (CDSS) have demonstrated significant potential in real-world applications for infectious disease management, particularly in addressing bacterial resistance.

The study assessed consistency by comparing CDSS recommendations with treatment decisions made by infectious disease specialists and non-specialists, using clinical data from bloodstream infection cases in healthcare institutions within a specific region. These data and analyses demonstrate the practical value of machine learning-based systems in infectious disease management. Such systems hold potential for enhancing early pathogen identification and optimizing antimicrobial treatment regimens, particularly improving the safety and rationality of antibiotic use in intensive care settings.

The paper features clear logic, well-organized methodology, transparent data and results, and explicit conclusions. There are no significant issues in the formulation of the scientific question, data analysis, or argumentation and discussion. It is already in a highly standard research paper format.

From a peer review perspective, if one were to be overly strict and nitpick, it might be possible to find some issues.

1. Data throughout the paper, tables and figures must maintain a consistent number of decimal places. While this may not pose a problem for researchers with some training, it could create unnecessary reading and comparison difficulties for cross-disciplinary readers—such as those engaged in interdisciplinary AI research within traditional biomedical fields, public health policymakers, patients, or the general public.

2. The truly appropriate conclusion for this paper is essentially the single sentence on lines 44–46. Or, to put it another way, lines 451–466, further condensed. Conversely, lines 468–474—the current conclusion of the paper—would be more suitable as an outlook within the discussion section rather than a conclusion directly derived from the research data presented.

.

Reviewer #1: No

Reviewer #2: No

---

## [Author Response · Author response to Decision Letter 1]

3 Jan 2026

RESPONSE TO ACADEMIC EDITOR

Comment 1: Missing References (Lines 62-64)

"The authors stated, 'Traditional diagnostic methods, such as blood cultures and phenotypic antimicrobial susceptibility testing (AST), remain the gold standard but suffer from long turnaround times, typically requiring 24 to 72 h for actionable results.' There are no references. More references should be cited."

Response: Thank you for this valuable observation. We have added appropriate references to support this statement (Lines 87): Reference 4 - 6

Comment 2: Funding Statement - Commercial Affiliation

"Please provide an amended Funding Statement declaring this commercial affiliation, as well as a statement regarding the Role of Funders in your study."

Response: We have amended the Funding Statement as requested. The updated statement is as follows (lines 639-643):

Funding Statement

The funder provided support in the form of salaries for authors [AF, JCGdlT, AR, MH-Z], but did not have any additional role in the study design, data collection and analysis, decision to publish, or preparation of the manuscript. The specific roles of these authors are articulated in the 'author contributions' section.

Comment 3: Competing Interests Statement

"Please also provide an updated Competing Interests Statement declaring this commercial affiliation along with any other relevant declarations relating to employment, consultancy, patents, products in development, or marketed products, etc."

Response: We have updated the Competing Interests Statement as follows (lines 654-664):

Competing Interests Statement

Ari Frenkel is Chief Science Officer of Arkstone Medical Solutions, the company that produces the OneChoice report evaluated in this study. JC Gómez de la Torre works as the Director of Molecular Informatics at Arkstone Medical Solutions and as the Medical Director at Roe Lab in Perú. Alicia Rendon and Miguel Hueda Zavaleta serve as Quality Assurance Managers at Arkstone Medical Solutions. These affiliations may be perceived as potential conflicts of interest. However, the study's design, data collection, analysis, interpretation, manuscript preparation, and the decision to publish the results were conducted independently, with no undue influence from the authors' affiliations or roles within the company. This does not alter our adherence to PLOS ONE policies on sharing data and materials.

Comment 4: Data Availability Statement

"When completing the data availability statement of the submission form, you indicated that you will make your data available on acceptance, in accordance with journal policies."

Response: We confirm that all data underlying the findings will be made freely accessible upon manuscript acceptance.

Comment 5: PLOS ONE Style Requirements

Response: We have carefully reviewed and ensured that the manuscript meets PLOS ONE's style requirements, including file naming conventions.

Comment 6: Reference List Review

Response: We have reviewed all references for accuracy and completeness. No retracted papers were cited. We have added new references as recommended and verified all existing citations.

RESPONSE TO REVIEWER 1

We sincerely thank Reviewer 1 for the thorough and constructive evaluation of our manuscript. We appreciate the recognition of our study's strengths, including the robust methodology, expert comparison design, quantitative rigor, and emphasis on antimicrobial stewardship. Below we address each concern raised.

Point 1: Narrow Clinical Focus

"While bacteremia is a serious and high-impact condition, the paper's exclusive focus on positive bloodstream infections means its findings apply only to a small minority of infectious disease patients... the real-world impact of CDSS in guiding therapy for the broader population of infectious diseases may be overstated."

Response: We appreciate this important observation and acknowledge the limitation. We have expanded the Discussion section to address this point.

Lines 562-565: We have added the following text:

"Second, our findings are specific to bacteremia and may not extend to non-bacteremic infections such as respiratory, urinary tract, or intra-abdominal syndromes. The real-world impact of this CDSS should be interpreted within the context of bloodstream infections."

Point 2: Scope of Clinical Decision Support

"Effective decision support in infectious diseases must transcend the management of positive blood cultures. The spectrum of infectious syndromes... requires nuanced recommendations that incorporate clinical presentation, local epidemiology, host factors, and dynamic changes in patient status."

Response: We agree with the reviewer's assessment and have added the following

Lines 607-610:

"However, future CDSS iterations must incorporate clinical presentation, local epidemiology, host factors, and dynamic patient status to provide comprehensive support across diverse infectious syndromes beyond bacteremia.”

Point 3: Outcome Measures

"The study centers on concordance rates rather than patient-centered outcomes, such as mortality, length of stay, or clinical improvement. While agreement with specialists is an important process metric, it does not guarantee improved patient outcomes."

Response: This is a valid and important criticism. We have acknowledged this limitation and highlighted the need for outcome-based studies.

Lines 566-569 (Added to Limitations):

"Third, we focused on concordance as a process metric rather than patient-centered outcomes such as mortality, length of stay, or clinical improvement. While substantial agreement with specialist judgment demonstrates clinical relevance, it does not directly prove improved patient outcomes."

Point 4: Generalizability

"The context—a Peruvian tertiary care setting—limits extrapolation to different healthcare systems, especially where diagnostic infrastructure, stewardship programs, and clinical workflows differ."

Response: We acknowledge this limitation and have added a statement addressing generalizability.

Lines 569-572:

“Fourth, findings from this Peruvian tertiary care setting may not generalize to healthcare systems with different diagnostic infrastructure, stewardship program maturity, or clinical workflows."

Point 5: Potential for Overreliance

"The risk remains that non-specialists may lean too heavily on AI recommendations without integrating broader clinical reasoning, especially in complex cases or where data inputs are incomplete."

Response: We appreciate this concern and have strengthened our discussion on this point.

Lines 572-574:

"Finally, lower concordance among non-specialists (κ = 0.61) highlights risks of overreliance on AI recommendations without adequate clinical reasoning, particularly in complex scenarios or when input data are incomplete."

Point 6: Regulatory Approval Omission (CRITICAL)

"A glaring omission in the paper is a clear statement regarding the regulatory approval of the OneChoice® software tool. Nowhere in the manuscript is it made explicit whether the CDSS is authorized as a medical device under major regulatory frameworks such as the U.S. Food and Drug Administration (FDA) or has received a CE mark under European legislation."

Response: We thank the reviewer for identifying this important omission. We have added a clear statement regarding the regulatory status of OneChoice®.

Lines 96-107 (Added to Introduction):

"However, OneChoice® is currently classified as a clinical decision support tool and has not yet received formal regulatory approval as a medical device from the U.S. Food and Drug Administration (FDA) or CE marking under European Union Medical Device Regulation. The system is intended to assist clinical decision-making and does not replace physician judgment. The recommendations generated should be interpreted within the context of individual patient characteristics and local guidelines, and the tool should be used under appropriate clinical supervision."

Recommendations for Future Research

The reviewer provided four recommendations for future research directions.

Response: We have incorporated these valuable suggestions into our revised manuscript.

Lines 582-598:

"AI-driven CDSS represent a promising advancement in infectious disease management. By enhancing diagnostic precision and standardizing antimicrobial therapy, these systems have the potential to mitigate the global burden of antimicrobial resistance and improve patient outcomes. However, future CDSS iterations must incorporate clinical presentation, local epidemiology, host factors, and dynamic patient status to provide comprehensive support across diverse infectious syndromes beyond bacteremia. Future research should prioritize prospective, multicenter studies assessing CDSS impact on patient-centered outcomes, antimicrobial resistance trends, and healthcare resource utilization across diverse settings. Key directions include: (1) expansion of CDSS validation beyond bacteremia to encompass respiratory, urinary, and intra-abdominal infections; (2) integration of dynamic clinical data streams, local epidemiological inputs, and rapid molecular diagnostics such as FilmArray® BCID2 to enable context-sensitive decision support; (3) incorporation of real-time resistance tracking and genomic insights to refine AI-CDSS applications; and (4) development of multidisciplinary implementation frameworks and training programs to ensure CDSS complements, rather than supplants, expert clinical judgment.”

RESPONSE TO REVIEWER 2

Point 1: Decimal Places Consistency

"Data throughout the paper, tables and figures must maintain a consistent number of decimal places. While this may not pose a problem for researchers with some training, it could create unnecessary reading and comparison difficulties for cross-disciplinary readers—such as those engaged in interdisciplinary AI research within traditional biomedical fields, public health policymakers, patients, or the general public."

Response: We appreciate this valuable observation regarding data presentation consistency. We have thoroughly reviewed all data presentations throughout the manuscript, tables, and figures to ensure uniform decimal place formatting.

Point 2: Conclusion

"The truly appropriate conclusion for this paper is essentially the single sentence on lines 44–46. Or, to put it another way, lines 451–466, further condensed. Conversely, lines 468–474—the current conclusion of the paper—would be more suitable as an outlook within the discussion section rather than a conclusion directly derived from the research data presented."

Response: We thank the reviewer for this insightful observation regarding the distinction between data-driven conclusions and future outlook statements. We have restructured the Conclusions section accordingly.

Changes Made (lines 601-612):

Conclusions

"This study demonstrates that the OneChoice® AI-based clinical decision support system achieves substantial concordance (κ = 0.70) with physician therapeutic decisions in bacteremia management, with particularly strong agreement among infectious disease specialists (κ = 0.78). The system showed a high overall concordance rate of 96.14% when considering any suggested treatment option, and 74.59% for the top recommendation. Infectious disease specialization was identified as the strongest predictor of concordance with CDSS recommendations (OR = 2.26, p = 0.001). The system demonstrated potential in reducing inappropriate antibiotic prescriptions, particularly unnecessary carbapenem use, supporting antimicrobial stewardship goals in a high-resistance setting. These findings support the utility of AI-CDSS tools as supervised decision aids for enhancing antimicrobial therapy standardization, especially in resource-limited healthcare settings."

We also added to end of Discussion (lines 582-598):

"AI-driven CDSS represent a promising advancement in infectious disease management. By enhancing diagnostic precision and standardizing antimicrobial therapy, these systems have the potential to mitigate the global burden of antimicrobial resistance and improve patient outcomes. However, future CDSS iterations must incorporate clinical presentation, local epidemiology, host factors, and dynamic patient status to provide comprehensive support across diverse infectious syndromes beyond bacteremia. Future research should prioritize prospective, multicenter studies assessing CDSS impact on patient-centered outcomes, antimicrobial resistance trends, and healthcare resource utilization across diverse settings. Key directions include: (1) expansion of CDSS validation beyond bacteremia to encompass respiratory, urinary, and intra-abdominal infections; (2) integration of dynamic clinical data streams, local epidemiological inputs, and rapid molecular diagnostics such as FilmArray® BCID2 to enable context-sensitive decision support; (3) incorporation of real-time resistance tracking and genomic insights to refine AI-CDSS applications; and (4) development of multidisciplinary implementation frameworks and training programs to ensure CDSS complements, rather than supplants, expert clinical judgment."

---

## [Editor Report · Decision Letter 1]

2 Feb 2026

Comparison of OneChoice® AI-based clinical decision support recommendations with infectious disease specialists and non-specialists for bacteremia treatment in Lima, Peru

PONE-D-25-43595R1

Dear Dr. Hueda-Zavaleta,

We’re pleased to inform you that your manuscript has been judged scientifically suitable for publication and will be formally accepted for publication once it meets all outstanding technical requirements.

Kind regards,

Benjamin M. Liu, MBBS, PhD, D(ABMM), MB(ASCP)

Academic Editor

PLOS One
---

## [Editor Report · Acceptance letter]

PONE-D-25-43595R1

PLOS One

Dear Dr. Hueda-Zavaleta,

I'm pleased to inform you that your manuscript has been deemed suitable for publication in PLOS One. Congratulations! Your manuscript is now being handed over to our production team.

Kind regards,

on behalf of

Dr. Benjamin M. Liu

Academic Editor

PLOS One